# Saliency-Guided Fine-Grained Temporal Mask Learning for Few-Shot Action Recognition

## ABSTRACT

Temporal relation modeling is one of the core aspects of few-shot action recognition. Most previous works mainly focus on temporal relation modeling based on coarse-level actions, without considering the atomic action details and fine-grained temporal information. This oversight represents a significant limitation in this task. Specifically, coarse-level temporal relation modeling can make the few-shot models overfit in high-discrepancy temporal context, and ignore the low-discrepancy but high-semantic relevance action details in the video. To address these issues, we propose a saliency-guided fine-grained temporal mask learning method that models the temporal atomic action relation for few-shot action recognition in a finer manner. First, to model the comprehensive temporal relations of video instances, we design a temporal mask learning architecture to automatically search for the best matching of each atomic action snippet. Next, to exploit the low-discrepancy atomic action features, we introduce a saliency-guided temporal mask module to adaptively locate and excavate the atomic action information. After that, the few-shot predictions can be obtained by feeding the embedded rich temporal-relation features to a common feature matcher. Extensive experimental results on standard datasets demonstrate our method's superior performance compared to existing state-of-the-art methods.

## CCS CONCEPTS

• **Computing methodologies → Activity recognition and understanding**.

## KEYWORDS

temporal relation modeling, few-shot action recognition, temporal mask learning

## 1 INTRODUCTION

Few-shot learning is able to generalize well to new unseen categories with only a few data samples and thus has gained increasing attention. It is promising to reduce the labor of collecting large-scale training data and the computational cost that many successful action recognition models [1, 11, 12, 40] will suffer when deployed in unfamiliar realistic scenarios where large-scale datasets are difficult to collect. However, video few-shot action recognition persists as a

*MM '24, October 28–November 1, 2024, Melbourne, Australia*
© 2024 Copyright held by the owner/author(s). Publication rights licensed to ACM.
ACM ISBN 978-1-4503-XXXX-X/18/06
https://doi.org/XXXXXXX.XXXXXXX

challenge, attributed to the intricate nature of action representation and temporal relationships.

Metric-based meta-learning methods [26, 28, 32] are one of the most widely used techniques in few-shot action recognition, where a meta-trained embedding network maps videos into a feature space and then matches the most similar video pairs in the feature space for recognition [3, 15, 41, 43, 44]. Most of the these methods obtain the similarity between videos by aggregating frame-to-frame similarity scores [4, 35, 36]. For example, OTAM [4] performs fixed time-order video frame alignment after obtaining video features to match the closest frame pair in two videos. HyRSM [36] devises a bidirectional mean Hausdorff metric to obtain frame-to-frame correspondences with aggregate matching relationships.

Despite the remarkable results, metric-based approaches to few-shot action recognition still exhibit limitations. Frame-level metric methods treat all frames in a video equally in the matching process, ignoring the fact that the semantic salience of each frame is different. Video frames can be classified into salient frames, non-salient frames, and ambiguous frames. However, current video models, trained on video-level semantic labels, tend to excessively prioritize salient frames that exhibit high dissimilarity to other support videos. Consequently, ambiguous frames may not receive adequate attention from the model, increasing the likelihood of false matching, especially when dealing with complex atomic actions [17] or those containing multiple semantic instances.

Recently, masked visual modeling has achieved stunning results in self-supervised learning of images and videos [13, 18, 20, 31], which masks out a part of the input and reconstructs the complete input with the remaining part, and it has been shown that masked visual modeling is capable of learning a more generalized feature potential space [18]. In addition, by designing a specific masking strategy, the model can be directed to focus on particular parts of the data. Building upon the aforementioned concepts, we posit that the introduction of mask feature reconstruction as a self-supervised pretext task into supervised few-shot learning for classification, which guides the model to perceive the semantics of the actions in the video more efficiently by reconstructing the video features, and improves the generalization ability of the few-shot action classification on the new task. However, in few-shot action recognition, indiscriminate escalation of the reconstruction difficulty may lead to an imbalance between the learning objectives of self-supervision and few-shot classification. Therefore, it remains a challenge to design a reasonable masking strategy so as to take full advantage of masked visual modeling to enhance the generalization ability of few-shot action recognition.

To address the above issues, we propose a novel saliency guided fine-grained temporal mask learning method for few-shot action recognition. Specifically, we built a temporal mask learning scheme in which fine-grained atomic action details are learned through interaction with the saliency-guided temporal mask. In general,

our study revolves around two key ideas. First, we learn a comprehensive temporal relation by forcing ambiguous frames to aggregate the complete video action semantics. Second, we exploit the low-discrepancy atomic action features to locate and excavate the atomic action information. Notably, to the best of our knowledge, our method is the first to learn the mid-level atomic action details compared to existing mask visual modeling work that typically uses the paradigm of reconstructing low-level features such as original pixels and HOGs [18, 37].

In summary, our contribution can be summarized as follows:

- we propose a novel saliency guided fine-grained temporal mask learning method that models the temporal atomic action relation for few-shot action recognition in a finer manner.
- We propose a temporal mask learning architecture to autonomously explore the optimal alignment for each atomic action snippet while incorporating a saliency-guided temporal mask module to adaptively locate and excavate the atomic action information.
- We conduct extensive experiments on five benchmark datasets to verify the effectiveness of the proposed method. The experimental results demonstrate the superior performance of our method compared to existing state-of-the-art methods.

## 2 RELATED WORKS

### 2.1 Few-Shot Action Recognition

Existing few-shot action recognition methods typically employ a metric-based meta-learning paradigm [32] and perform frame-to-frame temporal matching [4, 35, 36] to search for the most similar videos. Videos have an additional dimension of time than images, and thus the few-shot action recognition task requires a combination of both video spatio-temporal modeling and inter-video similarity metrics. For instance, AMeFu [15] incorporates the depth modality as supplementary scene information and fuses it with RGB modality for prototype metric [26]. OTAM [4] employs a variant of Dynamic Time Warping (DTW) to enforce the alignment of video frames in temporal order. ITANet [42] introduces a decomposed self-attention mechanism to alleviate intra-class variability in video features. TRX [24] utilizes CrossTransformers [9] to construct query-specific prototype representations. Building upon TRX [24], STRM [30] enhances local and global features to effectively capture spatiotemporal contextual information in videos. HCL [43] extracts multi-scale video representations from coarse to fine granularity, utilizing these representations for hierarchical matching. HyRSM [36], through task-aware video relation learning, tailors features specific to the task and employs a set-matching metric. MoLo [35] introduces a long-short contrastive loss to enforce local frame feature prediction with global context and perceives motion details through frame-wise difference reconstruction. SloshNet [39] adaptively integrates spatial features from different levels and integrates long-term and short-term temporal features for rich spatiotemporal characteristics. Our method focuses on providing discriminative video features, reducing potential ambiguities in frame-to-frame matching.

## 2.2 Mask Visual Modeling

Mask autoencoder is fundamentally a denoising autoencoder that learns effective feature representations by reconstructing the complete input from corrupted inputs. Recently, some work has applied Masked Image Modeling to self-supervised image pretraining, achieving remarkable results. For instance, iGPT [6] employs a self-supervised image pretraining approach by predicting the next pixel in a one-dimensional pixel sequence. BEiT [2] learns the visual semantics of images through predicting discrete visual tokens during pretraining. SimMIM [38] introduces a simple framework for the regression task on original image pixels. MAE [18] introduces an asymmetric encoder-decoder architecture and employs a high mask rate random masking strategy, enhancing the efficiency of masked image pretraining.

The tremendous success of Masked Image Modeling has sparked numerous efforts to extend this self-supervised pretraining paradigm to videos. BEVT [34] and VIMPAC [29] attempt a similar approach to BEiT [2], learning video representations by predicting features exported by a tokenizer. MaskFeat [37] sets the goal of mask reconstruction to HOG [7] handcrafted features instead of raw pixels. VideoMAE [31] and MAE-ST [13] use extremely high mask ratios to increase the reconstruction difficulty. VideoMAE [31] also proposes a tube masking strategy to mitigate information leakage during the reconstruction process, while VideoMAE v2 [33] introduces dual masking to explore larger video-based models, masking the input of the decoder to enhance training efficiency. MAR [25] introduces cell running masking, providing detailed context for the encoder to easily perceive the missing parts. In contrast to the aforementioned methods, our proposed saliency-guided fine-grained temporal mask learning is not aimed at achieving better pretraining performance but rather to assist in achieving more accurate video matching for few-shot action recognition.

## 3 METHOD

### 3.1 Problem Definition

The goal of few-shot action recognition is to obtain good generalization over new action categories with a small number of labeled samples. We follow the standard metric-based few-shot learning protocol [24, 32, 44] and employ the episode training paradigm. We sample videos from the meta-training set $C_{train}$ during training to construct multiple episodes, each of which consists of a set of support videos $S$ and a set of query videos $Q$. The support set S contains $N \times K$ samples from $N$ action categories, where each action category contains $K$ samples (called the $N$-way $K$-shot task), and the query set $Q$ contains several samples in N action categories. The training objective is to correctly recognize the video in $Q$ as one of the $N$ categories. During testing, episodes are built by sampling $N$ categories from the meta-test set $C_{test}$, where the action categories of $C_{train}$ and $C_{test}$ do not overlap with each other, in order to test the generalization performance of the model on unseen data.

### 3.2 Saliency-Guided Fine-Grained Temporal Mask Learning

**Overall architecture.** Figure 1 shows an overview of our framework. Our framework comprises two branches: the temporal mask

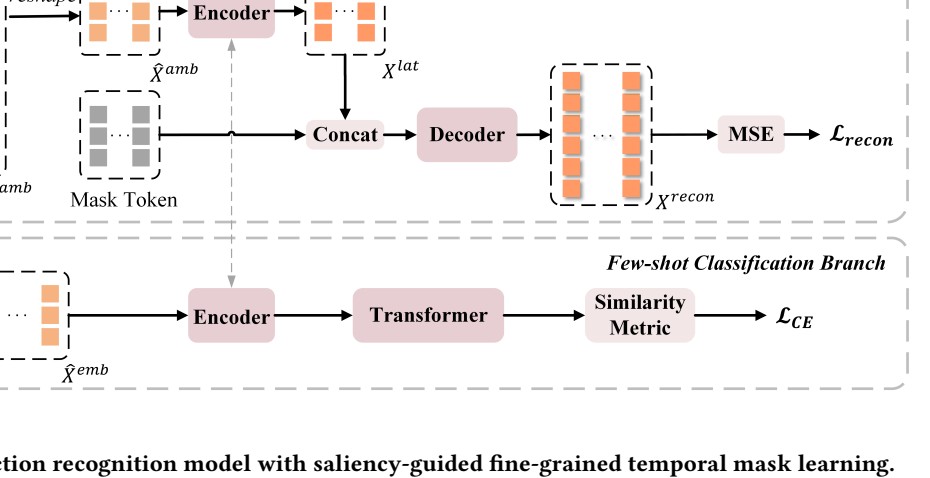

**Figure 1: Illustration of our few-shot action recognition model with saliency-guided fine-grained temporal mask learning.**

learning branch and the few-shot classification branch. In the temporal mask learning branch, we utilize the saliency-guided temporal mask generator module to adaptively generate a mask map for masking the feature embeddings, and then force the remaining portions to predict the features of the complete video. In the few-shot classification branch, we measure the similarity of action videos through frame-level matching for action classification. We aggregate the losses from both branches to form the overall framework loss.

In the episode under the N-way K-shot setting, the support set contains $N \times K$ videos, and the query set contains at least one video. For each video V, it can be represented as $V = \{I_1, I_2, \cdots, I_T\} \in \mathbb{R}^{T \times 3 \times H \times W}$, where $T$ is the number of video frames obtained by uniform sampling. We use a 2D CNN backbone network to extract frame-level features of the video V to obtain the feature embedding $X^{emb} = \{X_1^{emb}, X_2^{emb}, \cdots, X_T^{emb}\} \in \mathbb{R}^{T \times D}$. Each video frame is encoded as a feature vector in D dimensions.

**Saliency-guided temporal mask.** Given a frame-level embedded feature sequence $X^{emb}$ of a video, we obtain the category-aware scores for each frame using the following formula:

$$\mathcal{A} = \Phi_{cls}(X^{emb}) \tag{1}$$

Where $\Phi_{cls}$ is the composition of several temporal convolutional layers, interspersed with ReLU layers and Dropout layers. The resulting $\mathcal{A} \in \mathbb{R}^{T \times C}$ represents the semantic relevance between the frame-level features of the video and the class labels, where $C$ is the number of action categories in the meta-training set. Subsequently, a simple channel-wise summation operation ($\Phi_{sum}$) and a Sigmoid function are applied for category-agnostic aggregation, resulting in

the actionness score $\mathcal{A}^{ness} \in \mathbb{R}^T$. This operation can be expressed as follows:

$$\mathcal{A}^{ness} = Sigmoid(\Phi_{sum}(\mathcal{A})) \tag{2}$$

We consider the value of $\mathcal{A}^{ness}$ as the temporal saliency of all sampled frames in the video, and use it to guide the generation of the mask map. To excavate and learn more details about atomic actions. We focus on the ambiguous snippets (more detailed discussion in Section 4.3.2). We mask both salient and non-salient frames, compelling the network to aggregate comprehensive video semantics from ambiguous frames. Specifically, we perform top-k and bottom-k operations on the feature embedding $X^{emb}$ based on $A^{ness}$ to obtain two sets of frame features: $X^{sal} \in \mathbb{R}^{T_1 \times D}$ containing the top $T_1$ frames with the highest temporal saliency, and $X^{non} \in \mathbb{R}^{T_1 \times D}$ comprising the bottom $T_1$ frames with the lowest temporal saliency. The remaining frame-level features are considered as features corresponding to ambiguous frames, denoted as $X^{amb} \in \mathbb{R}^{T_2 \times D}$, where $T_2 = T - 2 \times T_1$. We mask $X^{sal}$ and $X^{non}$ and input only $X^{amb}$ as the visible portion into the encoder, relying on $X^{amb}$ to reconstruct the features of the entire video.

**Encoder.** Vision Transformer (ViT) [10] has demonstrated its powerful capability for global visual perception and is commonly employed as an encoder in Mask Visual Modeling methods [13, 18, 31]. Therefore, we choose shallow Transformer layers, similar to those in the ViT, as our encoder. In order to make the features extracted by the CNN suitable to be used as inputs to the Transformer block, we decompose the feature embedding into $N$ non-overlapping tokens, each of which has the shape $1 \times P$, i.e. $N = T_2 D/P$. We reshape $X^{amb} \in \mathbb{R}^{T_2 \times D}$ into $\hat{X}^{amb} \in \mathbb{R}^{N \times P}$, add positional embedding to identify the relative position of tokens,

and then feed them into the encoder to obtain the encoded latent features $X^{lat}$:

$$X^{lat} = \Phi_{enc}(\hat{X}^{amb} + PE) \tag{3}$$

$\Phi_{enc}$ refers to the encoder network, and $PE$ denotes positional embedding.

**Decoder.** Similar to the encoder, the decoder consists of shallow Transformer layers. We concatenate $X^{lat}$ with learnable mask tokens, add fixed positional embedding to indicate the position of each token in the video frame sequence, and utilize this as input for the decoder. Consequently, we obtain the reconstructed features $X^{recon} \in \mathbb{R}^{N' \times P}$, where $N' = TD/P$. The process can be expressed by the following equation:

$$X^{recon} = \Phi_{dec}(concat(X^{lat}, mask) + PE) \tag{4}$$

$\Phi_{dec}$ denotes the decoder network, while $mask$ represents mask tokens.

**Reconstruction loss.** To extract more semantic information from a small number of ambiguous frames, we introduce the reconstruction loss. The training objective of the temporal mask learning branch is to minimize the Mean Square Error (MSE) loss between the reconstructed and original features. We decompose $X^{emb} \in \mathbb{R}^{T \times D}$ into a token sequence $\hat{X}^{emb} \in \mathbb{R}^{N' \times P}$, where $P$ is the dimension of each token, and $N' = TD/P$ is the number of tokens, achieving the same size as $X^{recon}$, and then apply the MSE loss:

$$\mathcal{L}_{recon} = \frac{1}{N'} \sum_{t=1}^{N'} ||\hat{X}_t^{emb} - X_t^{recon}||_2 \tag{5}$$

where $|| \cdot ||2$ denotes the $\ell_2$ loss.

**Metric-based few-shot classification.** The frame-level feature sequence $X^{emb}$ undergo a notable refinement through the encoder of the aforementioned temporal mask learning branch. Nevertheless, a disparity persists between this refined representation and the requisite representation for the few-shot action recognition. As a result, in preparation for the frame-level metric procedure, we employ shallow Transformer layers [10] to further process the features, facilitating an improved alignment of these features within the metric space. The reshaped token sequence $\hat{X}^{emb}$, along with positional embeddings, is fed into the Transformer. The output are then reshaped into the size of $T \times D$, obtaining the features $X^{mat}$ for frame-level matching in videos. The above process can be represented by the following formula:

$$X^{mat} = reshape(\text{Transformer}(\Phi_{enc}(\hat{X}^{emb} + PE))) \tag{6}$$

$\Phi_{enc}$ represents the encoder in the temporal mask learning branch, Transformer stands for the shallow Transformer layers.

Given the features corresponding to the query video, denoted as $X_q^{mat}$, and the feature representations corresponding to videos in the support set $\mathcal{S}$ with the same labels as the query video, represented as $\hat{X}^{mat}$, the classification loss is expressed as follows:

$$\mathcal{L}_{CE} = -\log \frac{\exp(-\phi(X_q^{mat}, \hat{X}^{mat}))}{\sum_{s \in \mathcal{S}} \exp(-\phi(X_q^{mat}, X_s^{mat}))}. \tag{7}$$

$\phi$ is the distance metric function for video features, used to obtain video-level similarity by aggregating the similarity of frame-level features between support videos and the query video. The final loss can be expressed as:

$$\mathcal{L} = \mathcal{L}_{CE} + \lambda \mathcal{L}_{recon} \tag{8}$$

Our training strategy consists of two steps. Firstly, we freeze the feature embedding network and perform supervised training on $\Phi_{cls}$ using the class labels of video instances from the meta-training set. Subsequently, we freeze the parameters of $\Phi_{cls}$ and train our framework using the loss $\mathcal{L}$ from Equation 8. In the inference phase, given a query video $q$ from an unknown class and a support set $\mathcal{S}$, we use a similarity metric to find the video in $\mathcal{S}$ that is most similar to $q$. Subsequently, we assign the action class label of that video to $q$.

## 4 EXPERIMENT

### 4.1 Datasets and Experimental Setups

**Datasets.** We evaluate our approach on five standard datasets, including Kinetics [5], UCF101 [27], HMDB51 [27], SSv2-Full [16], and SSv2-Small [16]. The datasets are partitioned into meta-training, meta-validation, and meta-testing sets based on action categories to meet the requirements of the few-shot classification setting. For Kinetics, we follow the splitting strategy proposed by [44], selecting 100 action categories, each with 100 samples, and dividing these categories into 64, 12, and 24 for training, validation, and testing, respectively. For UCF101, we split it into 70, 10, and 21 categories for training, validation, and testing, following the same setup as [41]. In the case of HMDB51, we split it into 31, 10, and 10 categories for training, validation, and testing, adhering to the same splitting strategy as in [41]. For SSv2-Full and SSv2-Small, we adopt the split strategies utilized in [4] and [44], selecting 64 categories for training, 12 for validation, and 24 for testing from the datasets. The distinction lies in the fact that SSv2-Full comprises all samples for each category, whereas SSv2-Small only includes 100 samples per category.

**Implementation details.** Following the common paradigm of existing few-shot action recognition methods [4, 24, 35, 36], we employ ResNet50 [19] as the backbone network and initialize it with weights pre-trained on ImageNet [8] to extract frame-level features. We sparsely and uniformly sample 8 frames from each video, like previous methods [30, 35]. In the network architecture, the Transformer layers of the Encoder and Decoder are both configured with two layers. Additionally, we set k in the top-k and bottom-k, as well as $T_1$ and $T_2$ to 2, thus masking 4 frames. During training, we resize each frame in the video into $256 \times 256$, followed by random horizontal flips and random cropping to a $224 \times 224$ region. In the testing phase, we first perform resizing and then replace random cropping with center cropping. Similar to prior work [35], we collect 10,000 episodes from the meta-testing set to evaluate the model's performance and report the average accuracy. The weight parameter $\lambda$ in the loss function is set to 0.5. In Section 4.2, we employ two methods, OTAM [4] and Bi-MHM [36], as feature similarity metrics to train and evaluate our model. In Section 4.3, we exclusively utilize OTAM during the feature matching phase. We implement our framework using PyTorch [23] and conduct training on one RTX 4090 GPU.

**Table 1: Comparison with state-of-the-art few-shot action recognition methods on the Kinetics, UCF101, and HMDB51 datasets. Experiments are performed under the 5-way task with 1-shot, 3-shot, and 5-shot settings. The best results are denoted in bold black, the second-best results are indicated with an underscore, and "-" signifies that the result is not available in the published works.**

| Method | Reference | Kinetics | | | UCF101 | | | HMDB51 | | |
|---|---|---|---|---|---|---|---|---|---|---|
| | | 1-shot | 3-shot | 5-shot | 1-shot | 3-shot | 5-shot | 1-shot | 3-shot | 5-shot |
| CMN [44] | ECCV'18 | 57.3 | 72.5 | 76.0 | - | - | - | - | - | - |
| ARN [41] | ECCV'20 | 63.7 | - | 82.4 | 66.3 | - | 83.1 | 45.5 | - | 60.6 |
| OTAM [4] | CVPR'20 | 72.2 | 78.7 | 84.2 | 79.9 | 87.0 | 88.9 | 54.5 | 65.7 | 68.0 |
| AMeFu [15] | MM'20 | 74.1 | 84.3 | 86.8 | 85.1 | 93.1 | 95.5 | 60.2 | 71.5 | 75.5 |
| ITANet [42] | IJCAI'21 | 73.6 | - | 84.3 | - | - | - | - | - | - |
| TRX [24] | CVPR'21 | 63.6 | 81.8 | 85.9 | 78.2 | 92.4 | 96.1 | 53.1 | 66.8 | 75.6 |
| TA$^2$N [21] | AAAI'22 | 72.8 | - | 85.8 | 81.9 | - | 95.1 | 59.7 | - | 73.9 |
| STRM [30] | CVPR'22 | 62.9 | 81.1 | 86.7 | 80.5 | 92.7 | 96.9 | 52.3 | 67.4 | 77.3 |
| HyRSM [36] | CVPR'22 | 73.7 | 83.5 | 86.1 | 83.9 | 93.0 | 94.7 | 60.3 | 71.7 | 76.0 |
| Bi-MHM [36] | CVPR'22 | 72.3 | 81.1 | 84.5 | 81.7 | 88.2 | 89.3 | 58.3 | 67.1 | 69.0 |
| HCL [43] | ECCV'22 | 73.7 | 82.4 | 85.8 | 82.5 | 91.0 | 93.9 | 59.1 | 71.2 | 76.3 |
| Task Sampler [22] | MM'22 | 73.6 | - | 86.2 | 83.5 | - | 96.0 | 59.9 | - | 73.5 |
| SloshNet [39] | AAAI'23 | 70.4 | - | 87.0 | 86.0 | - | **97.1** | 59.4 | - | **77.5** |
| MoLo [35] | CVPR'23 | 74.0 | 83.7 | 85.6 | 86.0 | 93.5 | 95.5 | 60.8 | 72.0 | 77.4 |
| **Ours**+OTAM | - | 75.2 | 84.1 | 87.2 | **88.1** | 94.6 | 96.1 | 60.9 | 72.1 | 76.3 |
| **Ours**+Bi-MHM | - | **75.6** | **84.4** | **87.6** | 87.8 | **95.0** | 96.0 | **61.5** | **72.5** | 76.0 |

## 4.2 Comparison with state-of-the-art

In the 5-way task with 1-shot, 3-shot, and 5-shot settings, we compare our method with state-of-the-art approaches on Kinetics, UCF101 and HMDB51, as presented in Table 1. We utilize two commonly used frame-level alignment metrics as feature similarity metrics, namely OTAM [4] and Bi-MHM [36], to validate the effectiveness of our method. From the experimental results, it can be observed that when using Bi-MHM as the metric strategy, our method surpasses existing approaches across all three datasets in both the 1-shot and 3-shot settings. Specifically, in the 1-shot setting, our method achieves significant improvements of 1.5%, 1.8%, and 0.7% on Kinetics, UCF01, and HMDB51, respectively. Under the 3-shot setting, improvements of 0.1%, 1.5%, and 0.5% are observed on the same datasets, establishing a new state-of-the-art benchmark. When employing OTAM as the metric strategy, our method likewise demonstrates excellent performance outperforming numerous methods. Building upon Bi-MHM, Our method's performance in the 5-shot setting on the Kinetics dataset exceeds the previous best method by 0.6%. However, on UCF101 and HMDB51, it lags behind the best-performing methods. This problem is attributed to two possible reasons: (1) Methods such as STRM [30] and SloshNet [39], following the metric approach introduced in TRX [24], which is specifically designed for tasks with a higher number of shots, contributing to its superior performance in the 5-shot setting. (2) The 5-shot setting includes a higher number of samples in support set. Consequently, the number of discrepancy-based salient frames increases, while the number of ambiguous frames decreases. However, our method may obscure some information in salient frames, potentially leading to a decrease in accuracy. This problem is further elaborated in the subsequent experiments.

We further compare our approach with state-of-the-art methods on the SSv2-Small and SSv2-Full datasets, as presented in Table 2.

Our method achieves optimal performance under the 5-shot setting on both datasets, with a consistent improvement of 0.8%. However, under the 1-shot setting, our method slightly lags behind the best-performing approach, i.e. MoLo [35], which additionally leverages inter-frame difference information, thereby effectively enhancing performance on SSv2-Small and SSv2-Full. However, the framework of MoLo results in a larger parameter count (89.6M). In contrast, our approach achieves a parameter count (37.2M) only 0.42 times that of MoLo, striking a better balance between performance and efficiency.

Through analyzing the results of the first two experiments, we derive the following observations: For Kinetics, UCF101, and HMDB51, actions can be easily recognized through several key frames containing scene content, characterizing them as scene-centric datasets. On the other hand, the actions in SSv2 encompass complex temporal variations, classifying it as a temporal-centric dataset. For scene-centric datasets, having more key frames in the samples of support set makes it easier for the model to identify action category. For instance, in the 5-shot setting in Table 1, several previous methods achieve better results, whereas our approach performs better in low-shot settings. Conversely, in temporal-centric datasets, having more samples of support set results in more variable motion information, making it more challenging to learn. Our method effectively addresses this issue, yielding the best performance in the 5-shot setting. Therefore, in more complex temporal action matching scenarios, our approach demonstrates significant efficacy.

## 4.3 Ablation Study

*4.3.1 Impact of network components.* The results of the ablation study of network components are displayed in Table 3. Starting with a comparison of baseline and method 1, we add an Encoder to the baseline method, consisting of 2 Transformer layers without

**Table 2: Comparison with state-of-the-art few-shot action recognition methods on the SSv2-Small and SSv2-Full datasets. Experiments are performed under the 5-way task with 1-shot and 5-shot settings.**

| Method | Reference | SSv2-Small | | SSv2-Full | |
|---|---|---|---|---|---|
| | | 1-shot | 5-shot | 1-shot | 5-shot |
| MatchingNet [32] | NeurIPS'16 | 31.3 | 45.5 | - | - |
| MAML [14] | ICML'17 | 30.9 | 41.9 | - | - |
| CMN [44] | ECCV'18 | 33.4 | 46.5 | - | - |
| OTAM [4] | CVPR'20 | 36.4 | 48.0 | 42.8 | 52.3 |
| ITANet [42] | IJCAI'21 | 39.8 | 53.7 | 49.2 | 62.3 |
| TRX [24] | CVPR'21 | 36.0 | 56.7 | 42.0 | 64.6 |
| TA$^2$N [21] | AAAI'22 | - | - | 47.6 | 61.0 |
| STRM [30] | CVPR'22 | 37.1 | 55.3 | 43.1 | 68.1 |
| HyRSM [36] | CVPR'22 | 40.6 | 56.1 | 54.3 | 69.0 |
| Bi-MHM [36] | CVPR'22 | 38.0 | 48.9 | 44.6 | 56.0 |
| Task Sampler [22] | MM'22 | - | - | 47.1 | 61.6 |
| SloshNet [39] | AAAI'23 | - | - | 46.5 | 68.3 |
| MoLo [35] | CVPR'23 | **42.7** | 56.4 | **56.6** | 70.6 |
| **Ours**+OTAM | - | 40.0 | 56.0 | 54.1 | 69.8 |
| **Ours**+Bi-MHM | - | 41.3 | **57.5** | 55.4 | **71.4** |

**Table 3: Ablation study of three network components on Kinetics dataset under 5-way 1-shot and 5-way 5-shot settings. Encoder: shallow transformer layers; TML: Temporal Mask Learning; SGTM: Saliency-Guided Temporal Mask.**

| Method | Encoder | TML | SGTM | Kinetics | |
|---|---|---|---|---|---|
| | | | | 1-shot | 5-shot |
| Baseline | | | | 74.37 | 86.81 |
| 1 | ✓ | | | 74.50 | 86.87 |
| 2 | ✓ | ✓ | | 74.94 | 87.15 |
| Ours | ✓ | ✓ | ✓ | **75.18** | **87.18** |

mask reconstruction. We observe marginal performance improvements of 0.13% and 0.06% for 1-shot and 5-shot tasks, respectively, indicating that simply increasing the depth of the temporal feature extraction network does not lead to a significant performance improvement. Integrating temporal mask learning architecture (TML) into the aforementioned network for masked feature reconstruction, where the temporal mask is randomly generated, results in performance improvements of 0.57% and 0.34% for 1-shot and 5-shot tasks, respectively, demonstrating that the random temporal mask strategy can improve feature robustness to a certain extent. Subsequently, we further incorporate saliency-guided temporal mask module (SGTM). Compared to baseline that without temporal mask, we observe improvements of 0.81% and 0.37% for 1-shot and 5-shot tasks, respectively. Compared to random mask, SGTM can bring about more significant improvements. This suggests that selective temporal mask learning enables the network to selectively focus on learn more detailed actions, ultimately enhancing overall performance. Further elaboration on the learned action details will be provided in Section 4.4.

**Table 4: Comparison experiments on the performance of masking different regions on the Kinetics datasets. The regions masked by our method are the two frames with the highest temporal saliency and the two frames with the lowest temporal saliency.**

| Masking Region | Kinetics | |
|---|---|---|
| | 1-shot | 5-shot |
| 4 frames with the highest saliency | 75.01 | 86.30 |
| 4 frames with the lowest saliency | 74.99 | 86.49 |
| 4 frames with saliency in the middle | 74.88 | 86.34 |
| Ours | **75.18** | **87.18** |

**Table 5: Ablation study for the impact of different loss functions on the Kinetics and UCF101 datasets. $\lambda$ represents the weight of the reconstruction loss in Equation 8.**

| $\lambda$ | Kinetics (1-shot) | UCF101 (1-shot) |
|---|---|---|
| 0 | 75.05 | 87.81 |
| 0.1 | 75.13 | 88.06 |
| 0.3 | 75.08 | 88.09 |
| 0.5 | **75.18** | **88.10** |
| 0.7 | 74.85 | 87.77 |
| 0.9 | 74.83 | 87.58 |

*4.3.2 Analysis of different masking regions.* In our framework, we choose to mask frames with the highest and lowest temporal saliency. Here, we further explore the impact of masking different regions under the guidance of temporal saliency on the model's performance. Experiments are conducted on the Kinetics dataset under the 5-way 1-shot and 5-way 5-shot settings, and the results are summarized in Table 4. We observe that masking the top 4 frames with the highest temporal saliency achieves suboptimal performance in the 1-shot setting, masking the middle 4 frames results in the worst performance, and masking the bottom 4 frames yields 1-shot performance between the first two scenarios. In contrast, our method achieves the best performance in both 1-shot and 5-shot scenarios. The comparison results of different mask strategies in Table 4 indicate that:

- Masking salient frames can improve overall performance, suggesting that the network learns additional information from ambiguous frames and non-salient frames.
- Masking ambiguous frames leads to a performance decrease, indicating that non-salient frames can't provide sufficient information, causing the network to overly rely on salient frames.
- Our method's superior performance suggests that additional temporal information not present in the salient frames, which can be utilized for few-shot temporal matching and overall performance enhancement.

*4.3.3 Analysis of the impact of different loss functions.* We further investigate the impact of different loss functions on the performance of the method. Table 5 presents the performance variations of the method under different size of reconstruction loss on the Kinetics

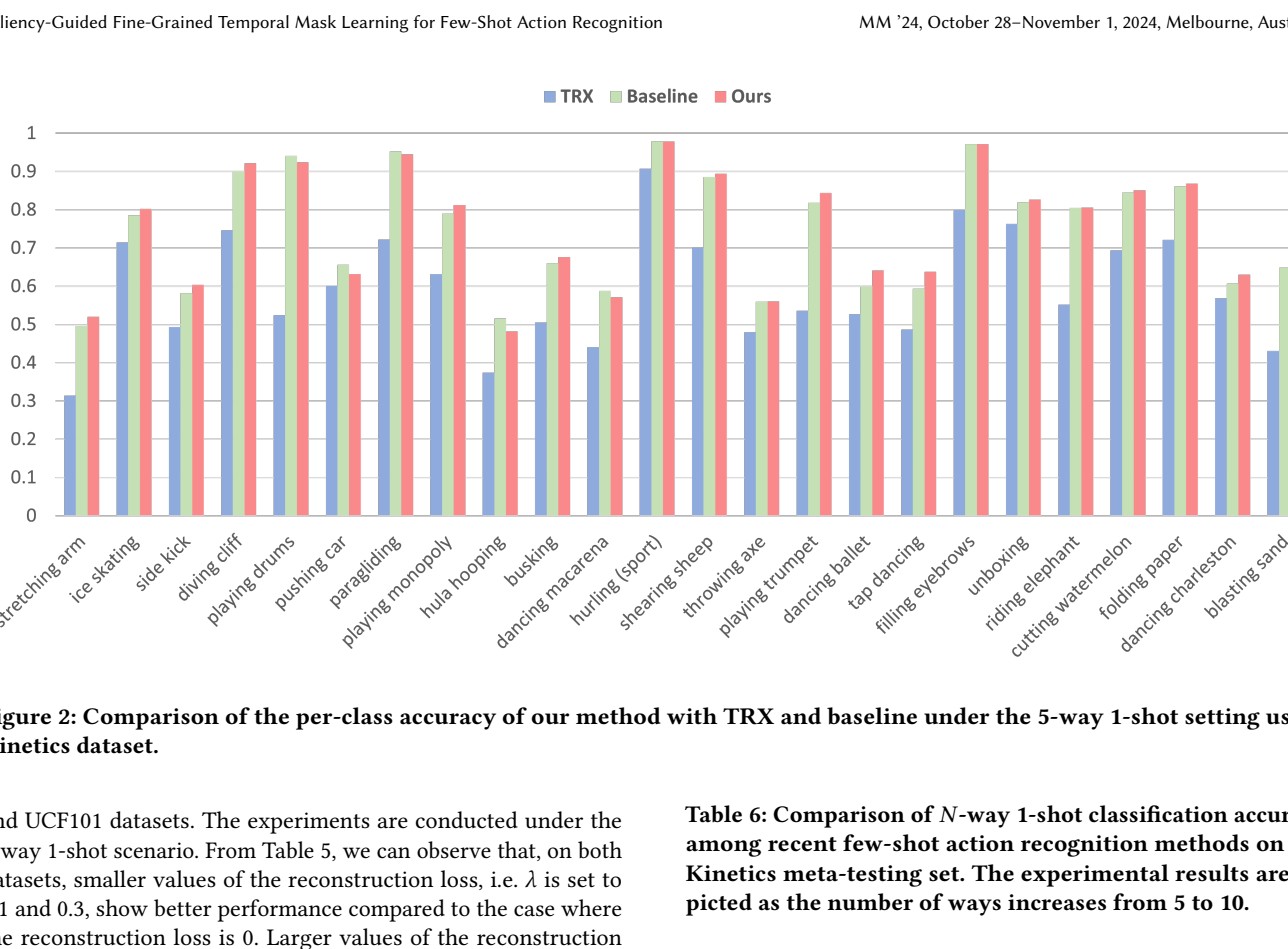

**Figure 2: Comparison of the per-class accuracy of our method with TRX and baseline under the 5-way 1-shot setting using Kinetics dataset.**

and UCF101 datasets. The experiments are conducted under the 5-way 1-shot scenario. From Table 5, we can observe that, on both datasets, smaller values of the reconstruction loss, i.e. $\lambda$ is set to 0.1 and 0.3, show better performance compared to the case where the reconstruction loss is 0. Larger values of the reconstruction loss, i.e., $\lambda$ set to 0.7 and 0.9, result in a performance decline. This is attributed to the stronger constraints imposed on the mask feature reconstruction network when the reconstruction loss is larger, enhancing the reconstruction capability of the encoder. The optimal performance is achieved when the value of the reconstruction loss is set between values mentioned aboves. Therefore, we set $\lambda$ to 0.5 as a balance point between the two losses to achieve the best performance.

*4.3.4 Analysis of per-class accuracy.* To investigate the impact of our proposed method on per-class classification accuracy for specific categories, we conduct a per-class accuracy analysis on the meta-testing set of the Kinetics dataset under the 5-way 1-shot setting. A comparison is made with TRX [24] and the Baseline, where the baseline comprises a ResNet50 backbone followed by 2 Transformer layers, and OTAM [4] is employed for metric learning. As depicted in Figure 2, in comparison to the baseline, our approach shows a reduction in accuracy for specific categories, such as "playing drums" and "pushing car". These categories involve interactions with large objects and lack distinct ambiguous segments. Consequently, by masking certain highly salient temporal segments, we experience a loss in performance. However, in comparison to our competitors, our method consistently demonstrates an overall performance advantage, which further highlights its applicability to the majority of motion patterns.

*4.3.5 N-way few-shot classification.* As shown in Table 6, we evaluate the accuracy of the model under the N-way 1-shot setting, where

**Table 6: Comparison of $N$-way 1-shot classification accuracy among recent few-shot action recognition methods on the Kinetics meta-testing set. The experimental results are depicted as the number of ways increases from 5 to 10.**

| Method | kinetics | | | | | |
|---|---|---|---|---|---|---|
| | 5-way | 6-way | 7-way | 8-way | 9-way | 10-way |
| OTAM [4] | 72.2 | 68.7 | 66.0 | 63.0 | 61.9 | 59.0 |
| TRX [24] | 63.6 | 59.4 | 56.7 | 54.6 | 53.2 | 51.1 |
| HyRSM [36] | 73.7 | 69.5 | 66.6 | 65.5 | 63.4 | 61.0 |
| MoLo [35] | 74.0 | 69.7 | 67.4 | 65.8 | 63.5 | 61.3 |
| **Ours** | **75.2** | **70.6** | **68.4** | **66.2** | **64.3** | **62.1** |

$N$ is incremented from 5 to 10, and compare it with some previous methods, including OTAM [4], TRX [24], HyRSM [36], and MoLo [35]. The experimental results show that the difficulty of few-shot classification continues to increase with increasing $N$, resulting in a gradual degradation of performance. Under a higher difficulty classification setting, i.e., the 10-way 1-shot classification setting, our method shows an accuracy advantage over the other methods, achieving an improvement of 0.8%, which further illustrates the effectiveness of the proposed method in dealing with difficult classification tasks. Meanwhile, our method consistently delivers the best results under all settings, demonstrating the excellent performance of the proposed method under different classification difficulties.

*4.3.6 Analysis of different masking ratio.* We further investigate the impact of varying masking ratios on model performance. Figure 4 presents the performance fluctuations of our model on Kinetics dataset with four different masking ratios, with experiments conducted under both 5-way 1-shot and 5-way 5-shot scenarios. From

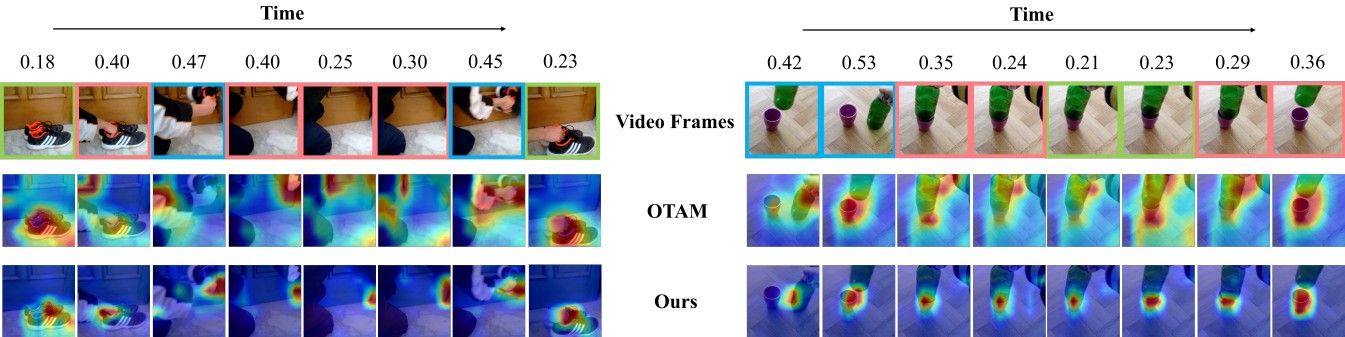

(a) Category: *Pretending to put something behind something*

(b) Category: *Failing to put something into something because something does not fit*

**Figure 3: Visualization of frame saliency and attention maps of our method and OTAM for two video instances on the SSv2-Full meta-testing set. Frames enclosed in the blue box are regarded as salient frames, those in the green box are non-salient, and the remaining frames in the red box are ambiguous.**

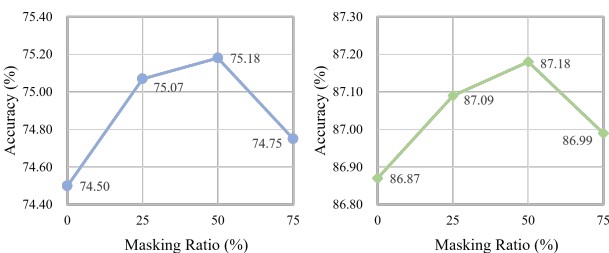

**Figure 4: Comparison experiments about the performance of different masking ratio on Kinetics datasets under 5-way 1-shot (left) and 5-way 5-shot (right) settings. In our method, we mask 50% of the frames within the video.**

Figure 4, we can observe that trend in performance variation with different masking ratios is consistent across both settings: it initially increases with the growth of the masking ratio, peaks at 50%, and then diminishes upon reaching 75%. This suggests that for the majority of video instances in the dataset, employing a 50% masking ratio effectively captures valuable information, while also learning more from previously overlooked information within ambiguous frames.

## 4.4 Visualization Results

To further investigate the impact of the saliency-guided fine-grained temporal mask learning strategy on the latent feature representation, we visualize frame saliency as well as attention maps for two action instances in the SSv2-Full meta-testing set. We observe and compare the learned feature content of our method with that of OTAM by studying the distribution of the spatial attention weights within the feature embedding network. As shown in Figure 3 (a), OTAM tends to focus on the correct regions in salient frames, but its attention is somewhat scattered on ambiguous frames, not concentrating on the regions where human limbs interact with objects.

This result in OTAM incorrectly interpreting the entire video sequence as the atomic action of "picking up shoes", failing to properly comprehend the atomic action of "placing shoes behind the door" contained in the ambiguous snippet. In contrast, our method consistently aligns its attention regions with the semantic regions of actions in the video, allowing it to correctly focus on the regions where human limbs interact with objects even in ambiguous frames.

From Figure 3 (b), it can be noted that, due to OTAM concentrating solely on differential information between frames within video instances, it exhibits attention divergence when there is no significant change in the video content (e.g. the third and fourth frames). It can also be observed that our method interprets actions based on interactions between objects rather than focusing on the objects themselves. This indicates that our method can accurately focus on the interaction between objects in the scene from a semantic perspective, which is particularly crucial in the SSv2 dataset where different objects interactions often occur in video instances. Such results validate the conclusion drawn in Section 4.3.1: Our saliency-guided temporal mask learning enables the network to learn more detailed semantic-level action features.

## 5 CONCLUSION

In this paper, we propose the saliency-guided fine-grained temporal mask learning method for few-shot action recognition. Our approach models the temporal atomic action relationships in videos through the temporal mask learning architecture, facilitating fine-grained matching for each atomic action segment. Additionally, we employ a saliency-guided temporal mask module to locate and excavate the fine-grained action snippet information. Extensive experiments demonstrate that our method effectively extracts temporal semantic information beyond the discrepancy-based salient frames. This enables a holistic understanding of action patterns from the perspective of overall motion, achieving a more comprehensive and refined modeling of temporal relationships. Consequently, our method exhibits excellent performance and holds a performance advantage over existing few-shot action recognition methods.

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
