# OpenReview forum: "Saliency-Guided Fine-Grained Temporal Mask Learning for Few-Shot Action Recognition"
_acmmm.org/ACMMM/2024/Conference — MM2024 Poster_

### Official Review · Reviewer_njzE · 2024-05-23

**Rating:** 4
**Confidence:** 3

**Summary:**

This paper proposes a saliency guided fine-grained temporal mask learning method for few-shot action recognition. The proposed approach employs a temporal mask learning architecture to model the temporal atomic action relationships in videos, and utilizes a saliency guided temporal mask generation module to generate mask maps, adaptively masking the features of salient and non-salient frames. By reconstructing the original video features from the features of ambiguous frames, it aims to extract temporal semantic information beyond salient frames, achieving more comprehensive and refined temporal relationship modeling. The experimental results verify the favorable performance of the proposed approach.

**Strengths:**

-This paper presents a novel exploration of applying mask visual modeling to few-shot action recognition. The authors leverage temporal saliency guidance of video frames and employ a temporal mask learning architecture to uncover often overlooked action relationships beyond the salient frames. This approach offers a new perspective and methodology for modeling temporal action relationships in few-shot action recognition.
-- The authors differentiate the frame sequences sampled from the video based on semantic saliency, with a particular focus on the low-discrepancy ambiguous frames. By extracting fine-grained temporal information from these frames, the representation of action features is enriched. Existing methods often overlook these ambiguous frames, whereas the authors provide a refreshing and effective insight in this regard.
-- In the section of method, the presentation of the problem definition provides readers with a clearer understanding of the issue. The method framework diagram has a clear flow and is easy to comprehend, the description of the method is well-structured.
-The proposed method exhibits excellent performance compared to existing state-of-the-art methods in multiple settings.
-The paper conducts comprehensive ablation studies, further validating the reliability and effectiveness of the proposed method. Additionally, the visualization results intuitively demonstrate the superiority of the method.
The article is clear in writing, easy to understand and follow.

**Limitations:**

-It is suggested to supplement the implementation details by specifying the specific dimensions of parameters such as $D$ and $P$ in the Section 3.2, and to clarify whether the Temporal Mask Learning Branch is inactive during the inference phase.
-The performance under the 5-way 5-shot setting on UCF101 and HMDB51 is limited.
-The abbreviations of modules such as TML and SGTM should be introduced earlier in the paper rather than being mentioned only in Section 4.3.
-In Figure 6, it is suggested to label "blue", "green" and "red" in the captions with their respective colors for better clarity and visual understanding.
-  The statements and explanations in Section 4.2 may be somewhat redundant. A more concise version is encouraged.

-  Some studies utilize multimodal frameworks for few-shot action recognition, such as CLIP-FSAR[1] and AMFAR[2]. Please include a discussion of these methods in Section2.1.

[1] Wang, Xiang, et al. "CLIP-guided prototype modulating for few-shot action recognition." International Journal of Computer Vision. 2023.
[2] Wanyan, Yuyang, et al. "Active exploration of multimodal complementarity for few-shot action recognition." Proceedings of the IEEE/CVF Conference on Computer Vision and Pattern Recognition. 2023.

**Suitability:**

3

---

### Official Review · Reviewer_hgX2 · 2024-05-24

**Rating:** 3
**Confidence:** 3

**Summary:**

This work focuses on few-shot action recognition task. Authors propose saliency-guided fine-grained temporal mask learning to help the temporal relation modeling，which introduces mask feature reconstruction as a self-supervised pretext task.

**Strengths:**

This paper explore the help of self-supervised tasks in few-shot action recognition tasks.
The experimental results demonstrate the effectiveness on the proposed method on several datasets (eg. kinetics, ucf)

**Limitations:**

——Why is the score obtained by formula (1) category-aware? Why can the subsequent actionness score reflect the salient or ambiguous nature of the frame? Lack of explanation and evidence.

——It can also be seen from the results in Table 4 that the differences between different masking strategies are actually very weak, and may just be experimental fluctuations. Neither intuitively nor empirically can explain the effectiveness of the saliency-guided mask proposed in the article.

——The related work in the paper also mentions work such as MaskFeat or VideoMAE. I think using their methods can also gain points. There is no strong evidence to convince me that the methods in this article are better than these. In addition, MoLo also used the self-supervised pretext task. They used decoder to reconstruct the frame difference and achieved better results on the SSv2 data set than this article.

——Line 342-345, I don’t understand why this kind of reshape is required. What is the difference between this and directly inputting the feature into the transformer?

——The main experiment table lacks some methods that need to be compared [1, 2]

[1] Khoi D Nguyen, Quoc-Huy Tran, Khoi Nguyen, Binh-Son Hua, and Rang Nguyen. 2022. Inductive and transductive few-shot video classification via appearance and temporal alignments. In European Conference on Computer Vision. Springer, 471 –487.

[2] Jiazheng Xing, Mengmeng Wang, Yudi Ruan, Bofan Chen, Yaowei Guo, Boyu Mu, Guang Dai, Jingdong Wang, and Yong Liu. 2023. Boosting Few-shot Action Recognition with Graph-guided Hybrid Matching. In Proceedings of the IEEE/CVF International Conference on Computer Vision. 1740–1750

**Suitability:**

3

---

### Official Review · Reviewer_pAP8 · 2024-05-24

**Rating:** 4
**Confidence:** 3

**Summary:**

This paper proposes a saliency-guided fine-grained temporal mask learning method that models the temporal atomic action relation for few-shot action recognition in a finer manner.

**Strengths:**

To model the comprehensive temporal relations of video instances, we design a temporal mask learning architecture to automatically search for the best matching of each atomic action snippet.

**Limitations:**

1. Please explain in detail how the saliency-guided temporal mask module is able to locate the features of keyframe without being guided by the loss function.
2. In the comparison results of the Table 1, why does the proposed method in this paper show sub-optimal results under the 5-shot setting on UCF101 and HMDB51 datasets.

**Suitability:**

3

---

### Official Review · Reviewer_4fM9 · 2024-05-25

**Rating:** 4
**Confidence:** 2

**Summary:**

This paper proposes a saliency-guided fine-grained temporal mask learning method that aims to model the temporal atomic action relation for few-shot action recognition in a more detailed manner. The effectiveness of the proposed method is demonstrated through experiments conducted on five datasets.

**Strengths:**

1. This paper proposes a novel saliency-guided fine-grained temporal mask learning method, which significantly improves performance when combined with OTAM and Bi-MHM.
2. The experimental analysis is thorough, providing a comprehensive evaluation of the strengths and weaknesses of the proposed method.
3. The writing is clear, providing detailed and comprehensive explanations. The figures and tables are visually intuitive.

**Limitations:**

1. The combination of OTAM and Bi-MHM in this paper leads to a noticeable improvement in performance. However, the proposed method does not achieve the best results on some datasets, which suggests the need to consider integration with more advanced models.
1. The method proposed in this paper is relatively simple and lacks significant innovation.

**Suitability:**

3

---

### Meta-Review · Area_Chair_qpnH · 2024-07-04

**Recommendation:** Accept (Poster)
**Confidence:** 4

**Metareview:**

This paper proposes a novel saliency-guided fine-grained temporal mask learning method to model the temporal atomic action relation for few-shot action recognition. The effectiveness of the proposed methods is demonstrated through experiments on five datasets. All the reviewers are inclined to accept. The AC agrees with the reviewers.

Quality: The  paper is well-structured, providing detailed and comprehensive explanations.

Clarity: The paper is easy to read, but some parts are not very clear.

Originality: The exploration of applying mask visual modeling to few-shot action recognition is interesting and effective.

Significance: Temporal modeling of atomic action relation is crucial for action recognition.